# Spatial Self-Phase Modulation in Graphene-Oxide Monolayer

**Tikaram Neupane** [1] , **Bagher Tabibi** [2] **, Wan-Joong Kim** [3] **and Felix Jaetae Seo** [2,*]

1    Department of Chemistry and Physics, The University of North Carolina at Pembroke,
     Pembroke, NC 28372, USA
2    Advanced Center for Laser Science and Spectroscopy, Department of Physics, Hampton University,
     Hampton, VA 23668, USA
3    K1 Solution R&D Center, Geumcheon-gu, Seoul 08591, Republic of Korea
*    Correspondence: jaetae.seo@hamptonu.edu

**Abstract:** The spatial self-phase modulation (SSPM) of the optical field revealed the magnitude and polarity of nonlinear refraction coefficients of the graphene-oxide (GO) atomic layers in an aqueous base solution with a resonant excitation using a chopped quasi-static laser at 532 nm. The SSPM of the optical field as a result of the intrinsic nonlinear refraction coefficient of GO atomic layers and the spatial distribution of intensity displayed the concentric diffraction rings at the far field due to the coherent superposition of transverse wave vectors. The number of concentric rings as a function of the applied intensity revealed the nonlinear refraction coefficient of GO which was estimated to be $\sim -6.65 \times 10^{-12}$ m$^2$/W for the laser-excitation duration of $\sim$0.32 s, where the negative polarity of nonlinear refraction coefficient was confirmed with the interference image profile of SSPM. The upper and vertical distortion of concentric rings at the far field at the longer laser-excitation duration of $\sim$0.8 s indicates the distortion of the coherent superposition of transverse wave vectors due to the localized thermal vortex of GO in the aqueous solution that offers novel platforms of thermal metrology based on localized optical nonlinearity and temperature-sensitive all-optical switching.

**Keywords:** spatial self-phase modulation; graphene-oxide atomic layer; nonlinear refraction





## 1. Introduction

The third-order optical nonlinearity of nonlinear absorption and refraction coefficients of the two-dimensional (2D) graphene oxide (GO) have intensively characterized for the photonic applications of pulse compression, mode-locking, Q-switching, optical limiting [1–5], and all-optical switching [6]. In addition to nonlinear optical applications of the GO, other several atomic layers were intensively studied for optoelectronic devices, biomedicine, bio-sensing, energy conversion, and storage [7–11].

The third-order optical nonlinearity can be characterized by various techniques including Z-scan, I-scan, four-wave-mixing, phase-modulation, etc. with either resonant or non-resonant excitation [1,12–16]. The third-order nonlinearity with the resonant excitation is a comparatively large and slow response than the non-resonant excitation [17]. The Z-scan and I-scan are simple techniques using a single beam, but Z-scan requires a reasonably long scanning time and I-scan includes a relatively high intensity which may hinder the characterization of highly absorptive optical materials. The FWM requires spatial overlap and temporal correspondence between pump-probe beams. The Z-scan reveals both the magnitude and the polarity of nonlinear absorption and refraction coefficients, and the I-scan and FWM characterize the cubic nonlinear susceptibility. The polarization-resolved forward-/backward-pump, probe, and signal beams in degenerate-/nondegenerate-FWM also reveal the physical origins of optical nonlinearity. The physical origins of optical nonlinearity may include the electronic transition [18], molecule reorientation or layer alignment, thermal effect, etc. The optical nonlinearity also has been characterized using spatial/temporal self-/cross-phase modulation for all–optical switches. The cross-phase

modulation has an optical time delay in all-optical switches which is not observed on monochromatic-based phase modulation or self-phase modulation [19–21]. Spatial self-phase modulation (SSPM) uses the radially distributed intensity of a Gaussian beam and the intrinsic nonlinear refraction coefficient of optical material. The coherent superposition of the transverse wave vectors, which have the phase difference due to the intrinsic nonlinearity and the spatial intensity distribution, display the concentric diffraction rings at the far-field [22]. The concentric diffraction rings are due to the constructive and destructive interferences depending on the even or odd integer $\pi$ nonlinear phase difference [23–25]. Therefore, the number of concentric rings as a function of the applied intensity reveals the nonlinear refraction coefficient of optical materials. The phase-modulation techniques recently have been applied to characterize the optical nonlinearity of GO atomic layers [6,26]. Shan and Xiang reported the third-order nonlinear susceptibility of GO dispersion using cross-phase modulation and described the thermal distortion of concentric diffraction rings [6]. Wang et al. disseminated the physical origin of diffraction patterns using different base solutions [26]. Further, Sadrolhosseini et al. [27] studied the nonlinear effect in Ag-NPs/GO and Au-NPs/GO nanocomposite using the spatial self-phase modulation technique at 532 and 405 nm wavelengths.

This article reports the polarity and magnitude of nonlinear refraction coefficients of the graphene oxide atomic layers in an aqueous solution using the SSPM technique with a resonant excitation, and the change of nonlinear refraction coefficient at the thermal vortex regions for the applications including the thermal metrology based on the localized optical nonlinearity and the temperature-sensitive all-optical switching.

## 2. Materials and Methods

The graphene oxide atomic layers in a base solution of deionized water were purchased from the Graphene laboratory [28]. The GO atomic layers in the aqueous solution include over 80% monolayers with ~0.5–5 μm lateral size. The optical width of GO atomic layers in an aqueous solution in the quartz cuvette was 10 mm for the cubic nonlinearity using the SSPM technique. The laser excitation source was a chopped CW laser at the peak wavelength of 532 nm at a frequency of 300 Hz. The laser beam with a diameter of ~1.6 mm was focused onto the GO atomic layers in an aqueous solution using a focal lens which has an effective focal length of 175 mm. The Rayleigh length and the beam waist at the focal point were ~1.1 mm and ~37 μm, respectively. The imaging screen of the diffraction ring was placed ~1.35 m away from the sample, and a CCD USB 2.0 image camera (DCU223C, Thorlabs, Inc, USA) monitored the diffraction image on the screen. The MATLAB code was utilized to convert the diffraction images to 2-D graphics for further analysis. In this work, the SSPM technique was focused to investigate the diffraction pattern along with laser interaction time and the number of rings with applied laser intensity to estimate the magnitude of nonlinear refraction coefficient and its thermal modulation simultaneously.

## 3. Result and Discussion

Figure 1 displayed the absorption spectrum of GO atomic layers in a base solution of DI water using a UV-Vis absorption spectrometer. The spectrum includes the UV absorption features at ~230 nm and ~300 nm wavelength and the visible absorption tail up to ~560 nm wavelength. The laser excitation spectrum with a peak at 532 nm for the SSPM technique is located at the edge of GO absorption.

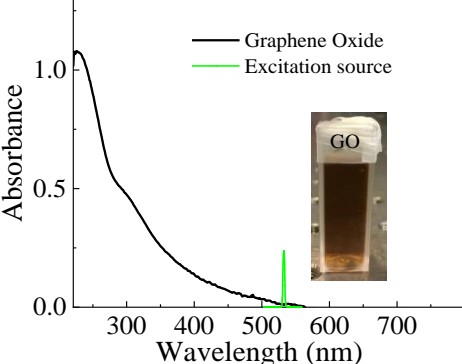

**Figure 1.** The absorption spectrum of graphene oxide atomic layers in the deionized water (DI) water (black), and the laser excitation source (green) at 532 nm for the SSPM.

The nonlinear refraction coefficient of graphene oxide atomic layers in the aqueous base solution was investigated using the SSPM technique with a resonant excitation at 532 nm. The transverse intensity distribution of the Gaussian beam and the intrinsic nonlinear refraction coefficient of GO atomic layers make the phase differences between the spatial optical fields of the Gaussian beam. The coherent superposition of the optical fields for the concentric rings at the far field depends on the even or odd integer number of $m\pi$ of nonlinear spatial phase shift $\Delta\phi_{NL} = \phi_{NL}(r_1) - \phi_{NL}(r_2)$ at the arbitrary positions $r_1$ and $r_2$ of a gaussian beam with the equal transverse propagation wave vector $\delta k(r)$. The maximum nonlinear phase shift $\Delta\phi_{NL}(0) = k\gamma I(0)L_{eff} \geq 2\pi$ is the primary condition of concentric rings at the far field, where $k$ is the wavenumber vector, $\gamma$ is the nonlinear refraction coefficient, $I(0)$ is the applied intensity, and $L_{eff} = \int_{L1}^{L2} \frac{1}{1+(z/z_o)^2} dz$ is the effective length of optical sample path $L$, where $L_1$ is the distance from the exit surface of sample to the focal point, $L_2$ is the distance from the input surface of a sample to the focal point, $L_2 - L_1 = L$ is the thickness of optical medium [24,29]. Therefore, the number of concentric rings ($N$) as a function of the applied intensity $I(0)$ reveals the nonlinear refraction coefficient $\gamma$ from the maximum nonlinear phase shift $\Delta\phi_{NL}(0) = k\gamma I(0)L_{eff} = 2\pi N$.

The typical images of SSPM diffraction rings of GO atomic-layer liquid suspension for the different excitation times are shown in Figure 2a. The base solution does not display the concentric rings even at the highest applied intensity within the experiment condition as shown in Figure 2b due to no nonlinear refraction from the base solution. Figure 2c is the quantitative analysis of the horizontal radius and the upper vertical radius of diffraction rings as a function of the laser excitation time at the applied intensity of ~48.8 MW/m$^2$, where the SSPM diffraction is proportional to the nonlinear phase shift, which is correlated to the nonlinear refraction and laser intensity [15]. The nonlinear refraction coefficient is the intrinsic property of the optical material, but it is not the case if the physical origin of nonlinearity changes. The analysis shows that the radius, including the number (not shown here), of concentric rings, is increased up to the ~0.32-s excitation time. It implies that the diffraction ring radius is increased as the net nonlinear refraction coefficient is increased due to the reduction of out-of-phase with collisions, impurity scattering, and boundary reflections which leads to the nonlocal electron (or charge carriers) coherence according to wind-chime model [30,31]. The distortions of upper vertical concentric rings from the graphene oxide atomic layer in an aqueous solution for the excitation times longer than ~0.32 s is due to the localized thermal vortex with heat convection [32–36].

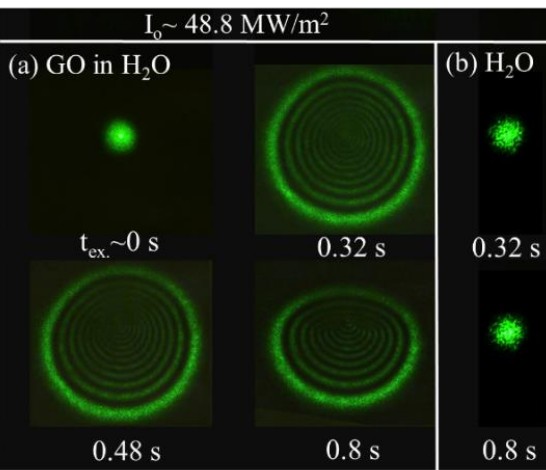

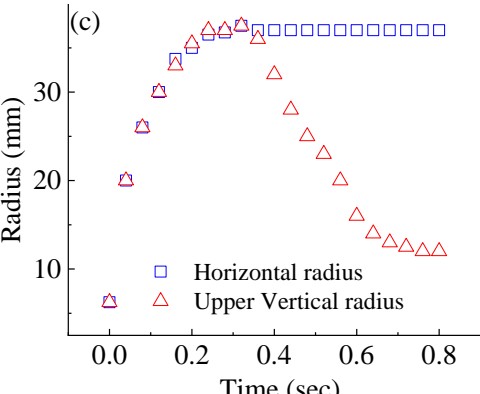

**Figure 2.** Typical SSPM diffraction patterns at the far field of a Gaussian beam (**a**) through the GO in aqueous base solution and (**b**) through the DI water for different laser excitation times, and (**c**) the horizontal and upper vertical radii of diffraction profiles as a function of laser excitation times. The applied intensity of $I_o$ ~48.8 MW/m$^2$ was used for (**a**–**c**). The data points for horizontal and vertical radii until 0.2 s are completely overlapped in (**c**).

The number of diffraction rings "$N$" increases linearly with the applied laser intensity, as shown in Figure 3, is described by [24],

$$N = \frac{\Delta\phi_{NL}(0)}{2\pi} = \frac{k\gamma L_{eff}}{2\pi} I(0)$$

The linear fitting slope, $k\gamma L_{eff}/2\pi$, of the number of $N$ as a function of applied intensity revealed the nonlinear refraction coefficient of GO. The coefficient was estimated to be ~$-6.65 \times 10^{-12}$ m$^2$/W [29,37] which has just an order difference from Shan's measurements of ~$3.57 \times 10^{-11}$ m$^2$/W using 532 nm and $1.1 \times 10^{-11}$ m$^2$/W using 671 nm CW laser source for few-layer GO [6]. In addition, the polarity of the nonlinear refraction coefficient was characterized by fitting with the Fraunhofer approximation of Fresnel–Kirchhoff diffraction integral to the concentric rings as shown in Figure 4a. The intensity distribution of Fraunhofer approximation at the far field is given by [38],

$$I = I_o \left| \int_0^\infty J_o(k_0 r\theta) exp\left(\frac{-r^2}{w^2(z)} - i\phi(r)\right) r dr \right|^2 \tag{1}$$

where $w(z) = w_o \sqrt{\left(1 + (z/z_o)^2\right)} = w_o \sqrt{R(z)/(R(z) - z)}$ is the beam radius at which the field amplitude falls to $1/e$ of their axial value, $k_0$ is the wave vector number, $\theta$ is the far-field diffraction angle, $Io = 4\pi^2 \left| \frac{E(0,z)exp(-\alpha L/2)}{i\lambda D} \right|^2$, $J_o(k_0 r\theta) = \frac{1}{2\pi} \int_0^{2\pi} exp(-ik_0 r\theta\, cos\varphi)d\varphi$ is the first kind of zero-order Bessel function, $E(0,z)$ is the applied field at the center of Gaussian beam, $\alpha$ is the linear absorption coefficients, $D$ is the distance between the sample and the screen, and $\phi(r)$ is the total phase shift which includes both linear and nonlinear phase shift [39]. Also, $R(z) = z\left(1 + (z_o/z)^2\right)$ is the radius of curvature and $w_o$ is the beam waist at the focal point, and $z_o$ is the Rayleigh length. The nonlinear phase shift $\phi_{NL}(z,r) = \Delta\phi_{NL}(0)exp\left(-\frac{2r^2}{w^2(z)}\right)$ includes the maximum nonlinear phase shift $\Delta\phi_{NL}(0) = k_s\gamma I_o L_{eff}$ [40]. Figure 4a displays the fitting (red color) and the experimental measurement (blue color) of concentric rings at the far field for the GO atomic layers in an aqueous solution which estimates the maximum nonlinear phase shift $(\Delta\phi_{NL}(0))$ $\sim -18\pi$. Figure 4b demonstrates the simulated diffraction pattern for the positive and negative maximum nonlinear phase shift to display the characteristic distinctions between the polarity of nonlinear refraction coefficient [41]. For the positive maximum nonlinear phase shift $(18\pi)$, the diffraction pattern unveiled a distinct central peak (blue) which was disappeared in the diffraction pattern (red) for the negative maximum nonlinear phase shift $(-18\pi)$. The suppression of the central peak indicates the negative nonlinearity or self-defocusing property of a given optical medium.

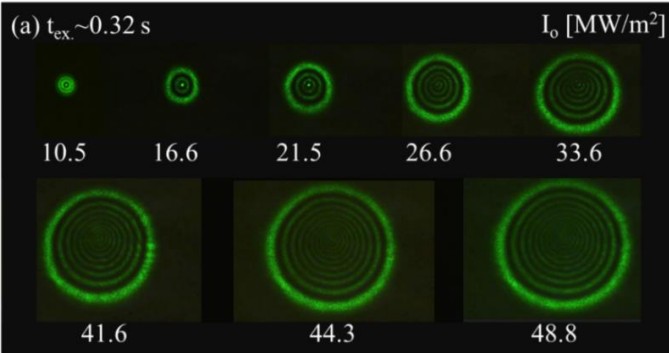

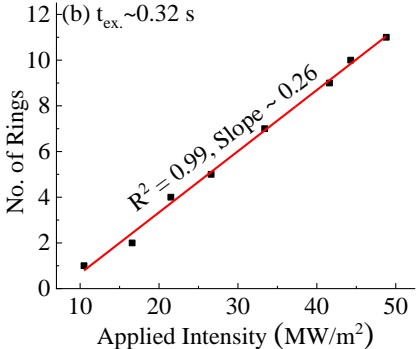

**Figure 3.** Typical SSPM diffraction patterns at the far field of a Gaussian beam (**a**) through the GO in aqueous base solution for different applied intensities [MW/m²] and (**b**) the number of diffraction rings as a function of applied intensity. The excitation time duration was $t_{ex.}$ ~0.32 s.

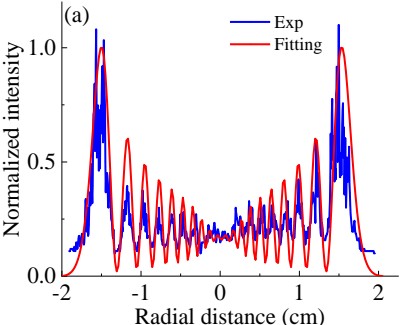 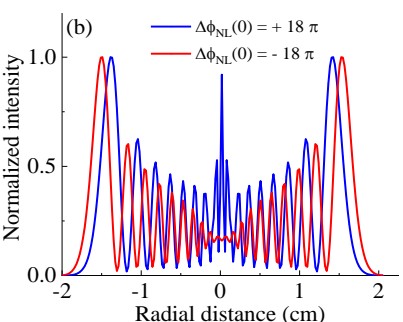

**Figure 4.** (**a**) Two-dimensional diffraction profile as a function of radial distance using a maximum nonlinear phase shift $\Delta\phi_{NL}(0) = -18\pi$ (**b**) Demonstration of diffraction patterns for positive and negative maximum nonlinear phase shift of $18\pi$.

Figure 5a is the schematic sketch of the half-cone angle ($\theta_H$) and the distortion angles ($\theta_D$) with respect to the diffraction profiles at the far field. The images of concentric rings for the different applied peak intensities at the excitation times of ~0.32 s and ~0.8 s are shown in Figure 5b. Also, the distortion angle to half-cone angle ratio ($\theta_D/\theta_H$) and its contribution to the nonlinear refraction due to the localized heat vortex as a function of applied peak intensity is quantitatively analyzed in Figure 5c. The distortion radius at the upper half of the diffraction profile is due to the laser-induced heat convection above the laser axis which results in non-axial asymmetrical isothermal in liquid base solution [40,41]. Vest and co-authors introduced heat convection due to a heating wire in liquid [42] which provides an analogy of the laser-induced heat convection of atomic layers in the base solution, and Wang and co-authors reported the laser-induced temperature fields [26]. Figure 5c indicates that the change of nonlinear refraction coefficient $\Delta\gamma = (\theta_D/\theta_H)|\gamma|$ at the localized thermal vortex due to the heat convection was estimated to be less than a half order in the unit of $m^2/W$ [37,43].

(a)

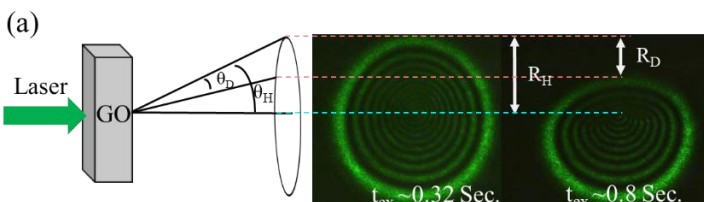

**Figure 5.** *Cont.*

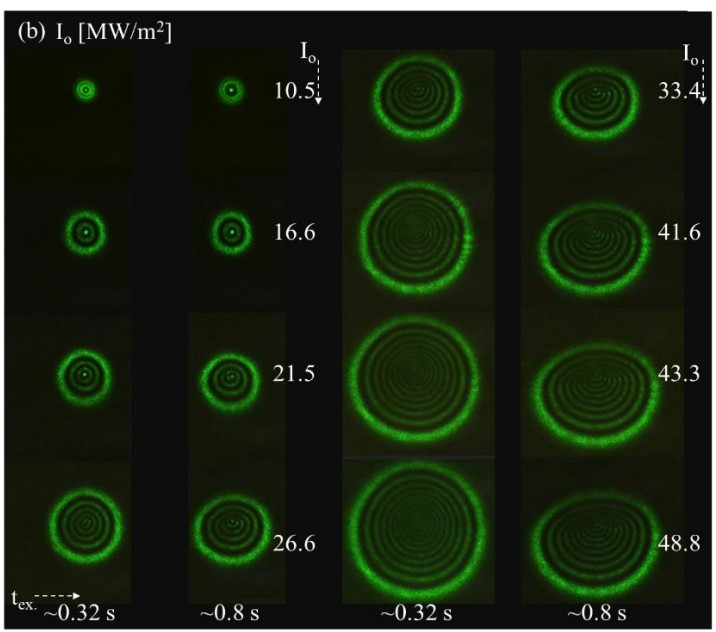

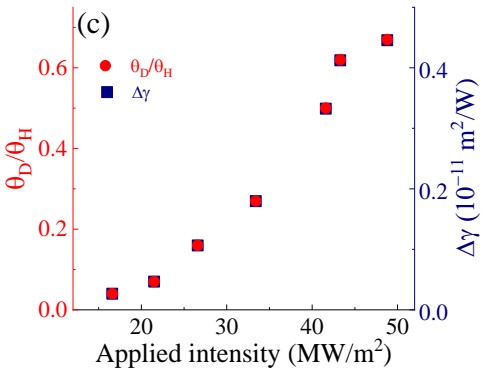

**Figure 5.** (**a**) Schematic sketch for the half-cone and distortion angles of SSPM, (**b**) SSPM diffraction images for different applied peak intensities at the excitation times of ~0.32 s and ~0.8 s, and (**c**) the ratio of $\theta_D/\theta_H$ (**left** y-axis) and the change of nonlinear refraction coefficient (**right** y-axis) as a function of applied peak intensity.

## 4. Conclusions

The nonlinear refraction coefficient of graphene oxide (GO) was estimated to be ~$-6.65 \times 10^{-12}$ m²/W using the SSPM technique with a chopped CW laser excitation at 532 nm and 300 Hz. The number of concentric rings as a function of applied intensity characterized the magnitude of the nonlinear refraction coefficient and fitting by the simulated Fraunhofer diffraction to the concentric rings revealed the polarity of the nonlinear refraction coefficient as well. The steady increase of the radius and the number of concentric rings below ~0.32 s excitation times implies the contribution of atomic layer alignment, according to the wind-chime model, and the thermal effect to the optical nonlinearity in addition to the electronic contribution at the absorption edge spectrum. The vertically asymmetric diffraction ring indicates the phase distortion of the optical field due to heat convection. The change of nonlinear refraction coefficient at the localized heat convex is around one-order in the unit of m²/W which may offer novel platforms of thermal metrology based on the localized optical nonlinearity and the temperature-sensitive all-optical switching.

**Author Contributions:** Conceptualization, F.J.S. and W.-J.K.; methodology, T.N.; formal analysis, T.N.; resources, F.J.S.; writing—original draft preparation, T.N.; writing—review and editing, F.J.S.,

B.T., T.N. and W.-J.K.; visualization, T.N.; supervision, F.J.S. and B.T.; funding acquisition, F.J.S. All authors have read and agreed to the published version of the manuscript.

**Funding:** This work at Hampton University was supported by NASA NNX15AQ03A and ARO W911NF-15-1-0535.

**Data Availability Statement:** The data are available from the authors upon reasonable request.

**Conflicts of Interest:** The authors declare no conflict of interest.

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
