# Peer review of "Spatial Self-Phase Modulation in Graphene-Oxide Monolayer"

_crystals, doi:10.3390/cryst13020271_

Round 1

Reviewer 1 Report

Comments to the Author:

Tikaram et al. study the spatial self-phase modulation of graphene-oxide (GO) atomic layers in an aqueous base solution with a resonant excitation using a chopped quasi-static laser at 532 nm and revealed the magnitude and polarity of nonlinear refraction coefficients of it. Authors confirmed the negative polarity of nonlinear refraction coefficient according to the interference image profile of SSPM. The change of nonlinear refraction coefficient at the thermal vortex regions gives the possible application of the thermal metrology based on the localized optical nonlinearity and the temperature-sensitive all-optical switching.

Though this work has obtained promising results, many aspects still need to be improved. Therefore, the following issues are recommended for further justification and clarification.

1.     As mentioned by authors, “the laser excitation spectrum with a peak at 532 nm for the SSPM is located at the edge of GO absorption”. Hence, whether authors have ability to do the SSPM with peak at other wavelength that may give different result of GO? In other words, whether the wavelength will affect the final analysis?

2.     Authors did good fitting process in this work but have not given enough analysis and explanation. For example, in figure 4a, authors achieve good fitting of two-dimensional diffraction profile as a function of radial distance by the intensity distribution of Fraunhofer approximation. However, in the experimental data, what does the vibration of normalized intensity mean? What will this fitting tell readers?

3.     Also, in the fitting process of figure 4, authors obtained nonlinear phase shift about -18π. Is it a common value when do this fitting? If not, what cause this parameter?

4.     Authors collect nonlinear refraction coefficient of graphene oxide as about 6.65×10-12 m2/W by using the SSPM technique. However, as far as referee known, there are also works research the nonlinear refraction coefficient of GO and other atomically thin layers. Authors are recommended to summarize the nonlinear refraction coefficient results from other works and do a comparation.

5.     Besides the research works containing parameters of nonlinear refraction coefficient. several papers will also help authors to analyze the applications of 2D materials. The papers with a recent study on 2D materials can be parts of the reference of this manuscript, such as Small 2022, 18, 2200016; Mater. Futures 2022, 1, 032302; Mater. Futures 2022, 1, 012301; Small 2022, 18, 2105383.

Reviewer 2 Report

The spatial self-phase modulation (SSPM) is a topic largely explored in different 2D materials. In this manuscript, Neupane et al. study in detail SSPM of Graphene-Oxide. Although this topic is already investigated by ref. 6, 18, the reported results generate an improvement in the knowledge of such effect.

The manuscript can be published after the authors fix the following issues:

1.       Youxian Shan, Jie Tang, Leiming Wu, Shunbin Lu, Xiaoyu Dai, Yuanjiang Xiang, Spatial self-phase modulation and all-optical switching of graphene oxide dispersions, Journal of Alloys and Compounds https://doi.org/10.1016/j.jallcom.2018.08.330 investigates the SSPM in graphene oxide. The authors should cite this work and they should take care about the other publication in which similar topic is involved.

2.      Line 71: The laser was chopped at 300 Hz. What is the role of chopping? What happens changing the frequency or removing the chopper?

3.      Fig. 2c:  In contrast with upper vertical radius, the value of horizontal radius is reported only for time longer than 0.2 s. The authors should explain this difference.

4.      The time evolution of SSPM is evaluated by the radius profile (fig. 2c), differently the intensity dependence is quantified by the number of rings (fig. 3). The authors should motivate the choice of two different quantification, otherwise they should report consistently radius ad number of rings for both time and light intensity.

Round 2

Reviewer 2 Report

see file
